# Preoperative and Intraoperative Methods of Parathyroid Gland Localization and the Diagnosis of Parathyroid Adenomas

**DOI:** 10.3390/molecules25071724

**Published:** 2020-04-09

**Authors:** Jacek Baj, Robert Sitarz, Marek Łokaj, Alicja Forma, Marcin Czeczelewski, Amr Maani, Gabriella Garruti

**Affiliations:** 1Chair and Department of Anatomy, Medical University of Lublin, 20-950 Lublin, Poland; robertsitarz@umlub.pl (R.S.); aforma@onet.pl (A.F.); amrmaanni@gmail.com (A.M.); 2Department of Surgery, Center of Oncology of the Lublin Region St. Jana z Dukli, 20-090 Lublin, Poland; m.lokaj@interia.eu; 3Chair and Department of Forensic Medicine, Medical University of Lublin, 20-950 Lublin, Poland; marcin.czeczelewski@gmail.com; 4Section of Endocrinology, Andrology and Metabolic Diseases, Department of Emergency and Organ Transplantations, University of Bari “Aldo Moro” Medical School, 70124 Bari, Italy; gabriella.garruti@uniba.it

**Keywords:** parathyroid gland, parathyroid adenoma, indocyanine green, carbon nanoparticles, autofluorescence, Raman spectroscopy, dynamic optical contrast imaging, laser speckle contrast imaging, shear wave elastography, imaging techniques

## Abstract

Accurate pre-operative determination of parathyroid glands localization is critical in the selection of minimally invasive parathyroidectomy as a surgical treatment approach in patients with primary hyperparathyroidism (PHPT). Its importance cannot be overemphasized as it helps to minimize the harmful side effects associated with damage to the parathyroid glands such as in hypocalcemia, severe hemorrhage or recurrent laryngeal nerve dysfunction. Preoperative and intraoperative methods decrease the incidence of mistakenly injuring the parathyroid glands and allow for the timely diagnosis of various abnormalities, including parathyroid adenomas. This article reviews 139 studies conducted between 1970 and 2020 (49 years). Studies that were reviewed focused on several techniques including application of carbon nanoparticles, carbon nanoparticles with technetium sestamibi (99m Tc-MIBI), Raman spectroscopy, near-infrared autofluorescence, dynamic optical contrast imaging, laser speckle contrast imaging, shear wave elastography, and indocyanine green to test their potential in providing proper parathyroid glands’ localization. Apart from reviewing the aforementioned techniques, this study focused on the applications that helped in the detection of parathyroid adenomas. Results suggest that applying all the reviewed techniques significantly improves the possibility of providing proper localization of parathyroid glands, and the application of indocyanine green has proven to be the ‘ideal’ approach for the diagnosis of parathyroid adenomas.

## 1. Introduction

Parathyroid glands (PGs) are small, nodular, endocrine structures that lie posterior to the thyroid. They are involved in the production of parathyroid hormone (PTH), which is a major hormone involved in the calcium homeostasis [1,2]. Any disruptions in PTH production might induce various parathyroid abnormalities such as primary hyperparathyroidism (PHPT), characterized by excessive PTH release [3]. The prevalence of PHPT is 1:1000 [4,5]. Impaired PTH production might result from various parathyroid abnormalities such as adenoma, cancer, hyperplasia or neoplasia [6,7]. Increased circulating levels of PTH may also be caused by parathyroid carcinoma (though in rare cases (1-2%)), but more often because of diffuse parathyroid hyperplasia (around 15%), and single or multiple autonomously functioning parathyroid adenomas (up to 85%) [8,9,10]. Thence, interrupted calcium homeostasis leads to further imbalances of other elements in the organism [11,12].

Parathyroid gland surgeries are hindered to some extent because of common issues with proper localization of the PGs. Extensive vascularization and vascular anastomoses near the PGs remain a common problem for the surgeons because of the risks of excessive intraoperative bleeding. Likewise, any damage to the PGs during surgeries can lead to more significant and long-term side effects such as hypoparathyroidism. Hypoparathyroidism can be permanent or transient, and typically resolves within 6 months [13]. Postoperative studies regarding hypocalcemia as a result of surgical procedures showed that transient hypocalcemia is observed in 27% of patients who underwent thyroidectomy, while permanent in 1% of cases [14]. Similarly, the risk of reoperation after incomplete removal of PGs during parathyroidectomies can reach even up to 30% (an upper limit); however, as it is affected by numerous factors, some centers report the risk to be relatively low (<5%) [15,16,17].

Another diagnostic problem with providing a proper localization of PGs is that due to their embryonic development, their anatomical position is not completely fixed. Usually, PGs are distributed on each side of the thyroid with the presence of two pairs in the embryonic development [18,19,20]. Embryologically, superior glands are developed from the fourth pharyngeal pouch, while inferior glands originate from the third pharyngeal pouch [21,22]. Eventually, superior glands are located on the superior border of the thyroid in close relationship with the recurrent laryngeal nerve and inferior thyroid artery. Because the distance traveled by inferior glands during embryological development is shorter, their location is more fixed compared to superior PGs [23].

Moreover, the anatomy of PGs varies among different populations and individuals. In the majority of cases, four PGs are present, nevertheless, there are individuals either with a less or excessive number of PGs (even up to twelve glands) [24]. The results of a recent meta-analysis, which involved 26 studies and 7005 patients, showed that 81.4% of patients had four PGs [25]. Regarding the geographical location, the results of the studies in North and South America revealed a smaller number of patients with four PGs compared to Europe. Among the studied groups, 4.9% and 6.3% of patients had five or more PGs, respectively.

Besides, there are cases in which not only additional but also ectopic PGs are present, leading to several surgical complications [26]. In such thyroid surgeries, when a broader area of lymph nodes must be dissected, PGs can be easily damaged. Therefore, it is fundamental for the surgeons to maximize the probability of localizing the PGs properly, to reduce the possible postoperative complications. 

Several preoperative examinations of the PGs, including cervical ECT scanning, ultrasonography (USG) or 99mTc sestamibi scintigraphy (MIBI) are commonly used; however, these methods are often not entirely sufficient for identifying the proper localization of PGs [27]. However, the effectiveness of providing a proper localization of PGs was significantly increased by some intraoperative methods, which appeared to be more frequently used [28]. Establishing the most exact localization of PGs seems to hold many promising results in reducing the number of surgical complications, including excessive bleeding or the accidental injury of the recurrent laryngeal nerve [29,30]. Intraoperative methods of PGs localization enable an intraoperative identification of the parathyroid abnormalities such as adenomas.

## 2. Materials and Methods

The authors have conducted a literature review of PubMed, Google Scholar, and Web of Science databases. The search used the following search string: (parathyroid gland OR parathyroid adenoma) AND (ultrasonography OR computed tomography OR sestamibi scintigraphy OR carbon nanoparticles OR Raman spectroscopy OR near-infrared autofluorescence OR autofluorescence spectroscopy OR autofluorescence imaging OR dynamic optical contrast imaging OR laser speckle contrast imaging OR shear wave elastography OR indocyanine green OR parathyroid autofluorescence OR parathyroidectomy). The literature search included both human and animal studies; also, there were no limits for the language or year of a publication. Additional relevant articles were obtained by the further literature search. Finally, 139 articles were included in this study (a time range from 1970 to 2020), among which those that were presenting a particular method of PGs or adenomas detection are summarized in Table 1.

## 3. Preoperative Imaging Techniques–Ultrasonography, Computed Tomography and Sestamibi Scintigraphy

Noninvasive preoperative imaging techniques, such as USG, MIBI, or computed tomography (CT) are commonly used while diagnosing pathologies within the thyroid region. Preoperative imaging techniques prevent unnecessary dissection and prolongation of surgery. Even though the preoperative imaging techniques significantly increase the surgical success as they facilitate localization of pathological PGs, they do not predict nor improve surgical outcomes (post-operative hypocalcemia or accidental parathyroidectomy). Among these, USG and MIBI are two modalities most frequently used for preoperative localization of parathyroid adenomas providing high sensitivity and specificity [31,32,33,34,35]. USG alone is usually efficient enough to localize parathyroid adenoma preoperatively; MIBI is proposed to patients with nonlocalizing USG (Figure 1) [36,37].

Besides, MIBI is more useful in localizing ectopic parathyroid adenomas, whereas USG – in intrathyroidal parathyroid adenomas [38]. Furthermore, there is an association between the utility of either USG or MIBI and the weight of a parathyroid adenoma [37]. However, Ozkaya et al. showed that correct identification of pathology is provided in 90.9% of cases while applying MIBI, and 87.1% with USG in patients with a single parathyroid adenoma [39]. Other studies shown the accuracy of preoperative localization of parathyroid adenomas using USG is higher (93%) comparing to MIBI (90%); the researchers have also showed that USG has higher sensitivity (98%) than MIBI (93%) [40]. A success rate of a unilateral approach for surgical excision of a parathyroid adenoma is 94.1% while providing preoperative localization by USG or MIBI [41]. Lee et al. showed that regarding preoperative localization of PGs, USG has the highest sensitivity (91.5%), while MIBI has the lowest (56.1%) [42].

Further, considering all three modalities, it was observed that the highest sensitivity was provided during USG and CT combination – 95.0%, whereas the lowest during MIBI and CT combination – 88.3%. Besides, a combination of all three modalities (USG, CT, and MIBI) presented with the highest sensitivity equal to 95.4%. Radioisotope techniques might not be sufficient enough in cases of double parathyroid adenomas or multiglandular hyperplasia [81,82]. Tokmak et al. suggest combining single-photon emission CT (SPECT-CT) with a dual-phase imaging method while searching for the localization of parathyroid adenomas preoperatively [83]. Thanseer et al. showed that ^18^F-fluorocholine (FCH) PET/CT has a higher accuracy (96.3%) comparing to USG or MIBI, and is effectively useful in localizing ectopic or small parathyroid lesions [38]. Furthermore, four-dimensional CT (4D-CT) provides significantly greater sensitivity (88%) for a precise (quadrant) localization of pathological PGs comparing to MIBI (65%), or USG (57%) [84]. Even though preoperative imaging techniques do not provide an identification of the majority of ectopic PGs, MIBI seems to be most useful in the identification of ectopic glands mainly in the thymus and upper mediastinal regions [85]. Despite high accuracy and sensitivity of preoperative imaging techniques, the need for intraoperative identification or verification of PGs is usually necessary.

## 4. Carbon Nanoparticles

### 4.1. Carbon Nanoparticles Characterization

Carbon nanoparticles were first invented and used as a tracer agent for imaging applications, especially for lymph nodes and vessel tracing [86]. These are nanosized carbon elements with a diameter usually less than 100 nm. One of the advantages of a carbon nanoparticle application is that they provide a clearer lymph node dissection region for surgeons without any toxic side effects on patients [87,88,89]. Carbon nanoparticles have a high affinity for the lymphatic system, promptness of dyeing black and the color contrast with tissues within the operative field. Thus, the carbon nanoparticles application is beneficial during open surgeries in the mediastinal region since it decreases the incident rate of PGs damage.

### 4.2. Carbon Nanoparticles in Parathyroid Glands Localization

The latest research confirmed the possibility of carbon nanoparticle application in PGs detection. In the following method, Shi et al. applied carbon nanoparticles in the suspension injection (1 mL/50 mg) within the area of the thyroid tissue [43]. The suspension contained nano-sized carbon particles with a diameter of approximately 150 nm. Carbon nanoparticles do not penetrate into the blood vessels because of their specific dimensions and are distributed in the lymphatic vessels and capillaries [90]. Because of this property, when the suspension is injected in the thyroid’s tissue region, it enters the lymphatic vessels specifically instead of the vascular vessels. During the surgery, injected carbon nanoparticles stain the thyroid and lymph nodes black, while leaving the PGs unstained enabling an easy differentiation between PGs and adjacent tissues.

While performing the preoperative procedure of applying carbon nanoparticles, it is crucial to provide a precise injection as possible. Thus, the injection should be done in the lower 1/3 of the ventral surface to each of the bilateral glands. The needle should not be inserted too deep or too shallow since it may cause extravasation of nanocarbon, which can eventually lead to the blackening of the surgical field.

The results of Shi et al. study showed that intraoperative application of carbon nanoparticles provided a 100% parathyroid detection rate. Furthermore, the rate of mistakenly cut PGs was much lower in the nanocarbon group (1.9%), comparing to the control group, which did not receive a carbon nanoparticle injection (15.6%). Moreover, the nanocarbon group presented a significantly lower hypoparathyroidism rate (19.2%) and postoperative hypocalcemia (13.4%), compared to the control group (42.2% and 22.2% respectively), because of the lower risk of intraoperative destruction of PGs while applying carbon nanoparticles during surgery. Shi et al. also showed that serum calcium and PTH levels were significantly higher in cohorts in which carbon nanoparticles were used compared to control groups [43]. Postoperative hypocalcemia requires the intraventricular supplementation of calcium - it significantly improves the clinical outcome of patients because of the sudden restoration of proper calcium concentrations.

The application of carbon nanoparticles seems to be a promising method for the intraoperative detection of PGs. This method enables PGs detection only by dyeing the surrounding tissues and not the PGs themselves. Therefore, differentiating between healthy PGs and adenomas is not possible with this method. Nevertheless, Yan et al. used carbon nanoparticles combined with ultrasound-guided fine needle aspiration and PTH rapid detection for diagnosis of parathyroid adenomas specifically [44]. With the guidance of ultrasound, PTH value was tested by the parathyroid puncture and further application of carbon nanoparticles was provided to check whether parathyroid adenomas would be stained black. The usage of this technique enabled a successful detection and diagnosis of all the parathyroid adenomas (n=12) in a studied group in a relatively short time (approximately 20-30 min). The sensitivity of this technique was also higher compared to USG, MRI, and MIBI where several detected parathyroid adenomas equaled to 6, 7, and 9 correspondingly.

The results of the available studies show that the average number of PGs detected during surgeries is significantly higher in groups where carbon nanoparticles are applied. Accurate detection of PGs by carbon nanoparticles provides better clinical outcomes, including a lower rate of hypocalcemia incidents, comparing to the control groups without carbon nanoparticles applied [91,92]. It is mainly because this method significantly reduces the risk of mistakenly cut PGs. Further, the blood supply can remain unchanged, preventing excessive intraoperative episodes of bleeding within the operated area. Due to the application of carbon nanoparticles as a possible method of unequivocal identification of PGs, the risk of postoperative hypothyroidism is significantly reduced [93,94,95]. Finally, the application of carbon nanoparticles provides more accurate localization of lymph nodes for further dissection during thyroidectomy and better visualization of metastatic lymph nodes [92,96,97,98,99,100].

## 5. Carbon Nanoparticles Suspension and Technetium Sestamibi (99mTc-MIB)

The study by Chen et al. sought to know whether the combined technique of nanocarbon particles and technetium sestamibi (99mTc-MIB) would provide a better prognosis for the localization of the PGs in comparison to 99mTc-MIB alone [45]. Intravenous injection of 99mTc-MiBI is a convenient method of identifying PGs. Recent studies have shown that small doses of technetium are as effective as higher doses, which minimize the amount of technetium needed during preoperative application and potential risks of its usage [101,102].

In the following study, two groups were considered: one group was treated with 99mTc-MIB alone, whereas the other with a combination of 99mTc-MIB with 0.1% carbon nanoparticles. Because of the presence of carbon nanoparticles, the thyroid gland and accompanying lymph nodes were stained dark blue, while the color of other tissues including PGs remained unchanged. Therefore, the overall duration of surgical procedures was significantly shorter. Additional studies confirmed that the application of carbon nanoparticles significantly facilitated the localization of pathological PGs, including hyperplastic or adenomatous glands [103,104].

As expected, the combination of carbon nanoparticles and preoperative suspension of 99mTc-MIBI, along with the radio guidance for the intraoperative localization of PGs enabled an easier and quicker detection compared to 99mTc-MiBI technique alone. The overall duration of the surgery was shorter for a group with carbon nanoparticles and 99mTc-MiBI combined, compared to 99mTc-MiBI alone (97 ± 16.6 vs. 115 ± 27.1 min, respectively, P = 0.015). Furthermore, a combination of carbon nanoparticles with 99mTc-MiBI provides an identification of PGs with synchronous localization, as well as those within the abnormal locations in the neck [102,105,106].

Therefore, it can be concluded, that carbon nanoparticles might significantly increase the effectiveness of localizing PGs. Nevertheless, further studies are needed to assess the utility of a combination of nanocarbon particles with 99mTc-MIB in establishing the localization of parathyroid adenomas specifically.

## 6. Raman Spectroscopy

Raman spectroscopy is a potential optical diagnostic technique that measures the inelastic scattering of light, which was described by the Nobel laureate Chandrasekhara V. Raman in 1928 [107]. After that, the Raman effect was particularly useful for medical applications because the scattered radiation measured is unique for each biomolecule [108]. Since that time, Raman spectroscopy has been applied in skin cancer margin assessment, diagnosis of endometriosis, differentiation between tumor and normal brain tissues, or between parathyroid adenomas and hyperplasia [46,109,110,111]. In this method, the frequency of a molecule, either vibrational or rotational can be changed with the use of laser light. This consequently produces a frequency shift, which afterwards is measured and analyzed. The results of this process are specific to the molecular constituents of the investigated sample.

### Raman Spectroscopy–Differentiation Between Parathyroid Adenomas and Hyperplasia

Primary hyperparathyroidism may be attributed to the solitary parathyroid adenoma [112]. Raman spectroscopy provides an accurate differentiation between two pathological conditions of PGs – either parathyroid adenomas or hyperplasia. The current standard for diagnosis of various pathologies of PGs is a histopathological study; nevertheless, it is crucial to diagnose such pathologies as early as possible for a better prognosis and clinical outcome. Since an intraoperative differentiation between benign or severe pathological conditions might be misleading, an accurate recognition of a specific pathology seems to be decisive during surgery.

Pathologically, parathyroid adenomas are oval or slightly round, lobulated and encapsulated tumors that appear greyish after section [113,114]. However, in many cases, parathyroid adenomas may also contain normal parathyroid tissue, which may rise further differentiation concerns. Raman spectroscopy allows for an accurate examination of the biochemical composition of a specific tissue. Das et al. have discussed applications of Raman spectroscopy which may significantly improve or even replace the need of intraoperative frozen sections for tissue pathology [47]. Palermo et al. showed that usage of Raman spectroscopy enables the possibility to distinguish between healthy and adenomatous parathyroid tissues, as the results of his study showed a 100% accuracy in an investigated sample of 18 patients [46]. Apart from distinguishing between adenomas and hyperplasia, Raman spectroscopy allows for specific identification of adenomatous histology – whether a parathyroid adenoma is primarily composed of a chief or oxyphil cells.

Even though Raman spectroscopy demonstrates high potential in differentiation between parathyroid adenoma and hyperplasia, further research is crucial to validate this methodology. One of the limitations of this method is the additional time needed to perform the analysis during surgery, which might extend its duration. Other disadvantages include additional costs and potentially higher post-operative side effects due to the probable prolongation of a surgery. Nonetheless, Raman spectroscopy seems to be a promising method of intraoperative differentiation between healthy and pathologically changed parathyroid tissues, as well as their further diagnosis [46,47].

## 7. Near-Infrared Autofluorescence

Parathyroid tissue possesses a characteristic of autofluorescence in the infrared wavelength spectrum [53,115]. Naturally present fluorophore of PGs is excited by 785-nm light and immediately emits longer wavelength light with a peak intensity at 822-nm [53,54]. The possible source of this property is vitamin D receptors or calcium-sensing receptors present in PGs cells [54,116]. Thyroid tissue shows many times lesser emission compared to PGs, whereas other tissues present in the operative site do not show autofluorescence. A study performed by McWade et al. showed that emission intensity from the PGs was lower in patients with high BMI, hyperparathyroidism, low vitamin D and high calcium levels [48]. It was independent of age, sex, ethnicity, and PTH level. The usage of near-infrared autofluorescence (NIR) in the localization of PGs has multiple advantages including no need for exogenous substances, instant feedback, non-invasiveness [49,50,51]. This method proved to facilitate the identification of PGs in 68% of investigated patients. Patients operated with NIR presented significantly lower post-operative hypocalcemia rates (5.2% to 20.9%), a higher mean number of identified PGs, and reduced autotransplantation rates. There was no difference in the rate of inadvertent resections [52,77].

Basing on the properties of PGs, several techniques were developed to provide their proper localization. One of the latest, original technique implemented near-infrared spectroscopy. The measurements were collected after the thyroid and PGs were exposed by placing the fiber optic probe to examined sites (S2000-FL^®^ fiber optic spectrometer Ocean Optics, Dunedin, FL, USA). The procedure required the darkening of the operation field [53,54]. An improvement of this method was a user-friendly clinical prototype of a device that uses near-infrared spectroscopy – PTeye [55,65]. It was adopted to perform measurements even with operating room lights turned on. Additionally, a foot-pedal activation method was implemented. All these factors cause that this device provides easily interpretable data and is more ergonomic for surgeons. However, near-infrared spectroscopy provides only the point measurements and requires contact with tissues, hence its value in localization of PGs is lower compared to near-infrared autofluorescence imaging. Falco et al. presented NIR imaging using a commercial device, the Fluobeam 800 laser (Fluoptics, Grenoble France) [56]. Other research groups used different either commercial or home-made devices [57,59,77]. The advantage of this technique is the ability to create a real-time image of a large part of the operation field without direct contact with tissues.

In 2016, Kim et al. performed a study that modified the technique of visualization of PGs called NIR-IR imaging [55]. The surrounding structures and surgical tools could be easily visible because of the additional infrared illumination in broadband (700-1400 nm). Thus, this method provides the spatial context and enhances the probability of determining a proper localization of PGs in the surgical field. Further development of NIR was presented in 2018 by the Vanderbilt research group. The overlying tissue imaging system can display the results of autofluorescence measurements directly on the operation field. The strong green light marks PGs and supports the surgeon’s visual interpretation. This method excludes the need of looking at external monitors and issues related to the inappropriate interpretation of images from the screen [15]. To best of our knowledge, only one study focused on the identification of primary hyperparathyroidism with NIR. Kose et al., investigated the utility of near-infrared autofluorescence imaging (Fluobeam device) in the identification of hyperfunctioning PGs [60]. On a sample of 199 PGs, autofluorescence was detected in 192 cases. Due to abnormal visual appearance, 77 glands were excised, and 65 appeared to be hyperfunctioning according to Miami criteria. Eventually, hyperfunctioning PGs showed lower autofluorescence and its heterogeneous pattern compared to normal functioning glands. Further research has confirmed the utility of NIR imaging to confirm the presence of parathyroid tissue within surgical specimens [61]. In 2019, a study by Henegan et al. presented a case of intrathyroidal parathyroid adenoma, which was detected via NIR imaging [62].

The application of NIR does not provide information about the viability of the PGs since the fluorescence persists after devascularization. Hence, Alesina et al. proposed combining the autofluorescence with the application of indocyanine green to assess PGs vascularity during the resection [63]. Another disadvantage of NIR is the necessity of turning the operating room light completely off to register the fluorescence, which disrupts the operating room workflow and extends the operation time. So far, the only method insensitive to light is the PTeye device [55]. Furthermore, NIR shows a limited ability to localize PGs covered deeply by other tissues since the penetration of light is a few millimeters of soft fatty-fibrous tissue [58,64].

## 8. Dynamic Optical Contrast Imaging

PGs detection has been significantly improved in 2017 by a novel method presented in a study performed by Kim et al. – dynamic optical contrast imaging (DOCI) [66]. In this method, tissues are illuminated by LEDs with different wavelengths and fluorescence decay information from endogenous fluorophores is used to obtain pixel values and create color diagrams [117]. In the ex vivo study, the 127 specimens were examined by macroscopic characteristics, DOCI maps, and histopathological images. DOCI maps showed differences in the relative decay between parathyroid and fat, thymus as well as thyroid tissues in all examined wavelengths (407-676 nm). The wavelengths within the range of 465-594 nm showed the highest contrast between thymus and parathyroid tissue. Even promising results of ex vivo studies, DOCI needs to be used in in vivo studies to establish it as a method of intraoperative PGs detection.

## 9. Laser Speckle Contrast Imaging

Mannoh et al. used a laser speckle contrast imaging (LSCI) to evaluate PGs viability during the thyroidectomies and parathyroidectomies [67]. It is a real-time, contrast-free, objective technique that allows distinguishing vascularized and compromised PGs. LSCI provides the detection of the movement of particles such as red blood cells, a few hundred microns beneath the tissue surface [118,119]. It is possible due to differences in light scattering, which produce a pattern of bright and dark areas depending on the velocity of particles. The obtained pattern is called a speckle pattern. The LSCI was tested on 20 patients with thyroidectomy and compared to a gold standard - surgeon’s visual assessment of the gland’s viability. Viable PGs had lower speckle contrast compared to compromised ones. Using statistical methods (Receiver operating characteristic) the threshold to distinguish both groups was set up to 0.09, which gave 92.6% sensitivity and 90.6% specificity with total accuracy 91,5%. The validation was conducted on 8 patients undergoing parathyroidectomies. The advantage of this method is relatively fast and specific identification of damaged PGs compared to the technique of PTH hormone assay, which gives results after 10 min and does not specify which gland is devascularized. One disadvantage of LSCI is its susceptibility to any kind of movement of an operation field, including the patient’s breathing or surgeon’s movement. Yet, in the aforementioned study, the researchers were able to obtain distinct differences in speckle contrast in the trial.

## 10. Shear Wave Elastography

Various imaging techniques such as sonography, MIBI, CT, or MRI are used on regular bases to evaluate parathyroid lesions, however, in some cases, their diagnostic accuracy is limited [120,121,122]. Differences in the structure of PGs and adenomatous glands can be estimated by the variations of gland stiffness – parathyroid adenomas are stiffer because of the reduced fat tissue within the pathologically changed glands. The presence of a fibrous, hard capsule around parathyroid adenoma also increases its stiffness. Furthermore, parathyroid adenomas are stiffer comparing to hyperplasia or adjacent lymph nodes. This characteristic was applied in the shear-wave elastography (SWE), which is a potential technique providing measurements of PGs stiffness [123].

Hattapoğlu et al. showed that among 36 examined patients, the mean shear wave velocity (SWV) ± SD of parathyroid hyperplasia lesions (n = 4) was 1.46 ± 0.23 m/s, whereas, in case of parathyroid adenomas (n = 32), SWV was 2.28 ± 0.50 m/s [68]. In a study, a significant difference was shown between normal thyroid tissue and parathyroid adenoma (*P* < 0.001). Further, a difference between thyroid parenchyma and thyroid nodules was also significant (*P* < 0.001). However, no significant difference was shown between thyroid nodules and parathyroid adenomas (*P* = 0.989). Even though it was the first time of SWE application in the identification and differentiation of parathyroid lesions such as adenomas, the results turned out to be credible and promising. Azizi et al. showed that parathyroid adenomas and thyroid tissue present a statistically significant difference in SWE (*P* < 0.0001); according to researchers, SWV measurements also enhances other preoperative sonographic parameters, providing a more accurate diagnosis of parathyroid adenomas [69].

Application of SWE with Virtual Touch tissue imaging quantification (VTIQ) enabled the differentiation between parathyroid adenoma and hyperplasia, as well as cervical lymph nodes [124]. Golu et al. suggested that the cutoff value for SWE-Mean should be set at 12.5 kPa to provide an accurate diagnosis of parathyroid adenoma [70]. According to researchers, subjects with SWE-Mean value lower than 12.5 kPa present a high probability of parathyroid adenoma presence. It was proposed that SWE alone would provide sufficient diagnosis without a need for any additional imagistic tests. Contrarily, Stangierski et al. claim that SWE should constitute only an additional preliminary diagnostic tool for parathyroid adenoma identification [71]. SWE combined with acoustic radiation force impulse (ARFI) technology can be applied to increase the diagnostic accuracy of USG in the identification of parathyroid lesions primarily in patients with hyperparathyroidism [72]. Furthermore, SWE with ARFI enables the differentiation between parathyroid adenomas and benign or malignant thyroid nodules [73]. One limitation of SWE is that dimensions of parathyroid adenomas cannot be estimated, however, according to recent studies, SWE seems to be a valuable, additional technique providing an accurate parathyroid adenoma identification.

## 11. Indocyanine Green

Indocyanine green (ICG) is a near-infrared fluorescent agent that has been used since the 1970s, mainly in retinal angiography [125]. ICG is a non-toxic, inert, organic compound that is injected intravenously and binds to plasma proteins, which eventually become illuminated with the usage of a low-energy laser at 806 nm. This property enables a charge-coupled device camera to record the fluorescence of ICG molecules [78]. Thus, this characteristic has been applied in the visualization of PGs since the fluorescence intensity of the glands can be measured by the ICG fluorescence angiography (ICGA) [74,126,127]. ICG is a nonselective agent, which constitutes a limitation of its intraoperative application in PGs detection since it does not target parathyroid parenchyma specifically. However, because PGs receive a higher amount of blood compared to surrounding tissues, they emit a much stronger fluorescent signal, which consequently presents the exact localization of the PGs (Figure 2) [128].

ICG, at the very beginning (starting in 2015), has been used as a way of detecting PGs in dogs [129]. Further studies reported that ICG can apply to humans in many broad specialties like cholangiography, perfusion assessment of gastrointestinal anastomoses, adrenalectomy, or real-time lymph node mapping [79]. Further, ICG can be used in assessing blood perfusion within the PGs, but also in other cases including skin flaps, bowel anastomosis or lower limbs [126]. Several studies reported that ICG can be used in the intraoperative localization of parathyroid adenomas. The first successful determination of a parathyroid adenoma with the usage of ICG was performed in 2015 by Chakedis et al. [80]. In this study, a previously performed sestamibi scan provided false negative results since it did not detect the parathyroid adenoma. Contrarily, a CT scan which was also performed detected a heterogeneous soft tissue mass in the region of the thyroid lobe. In this case, 2.5 mg/mL bolus of 3 mL of ICG was injected, and the fluorescence was presented on the neck approximately just 20 s after injection. Nerve monitoring provided a successful surgical approach, causing no harm to the recurrent laryngeal nerve.

In a similar study performed by DeLong et al., it has also been shown that among 54 patients who have a sestamibi scan performed, 36 of them presented a parathyroid adenoma, whereas 18 were not detected, even though the adenomas were present [79]. However, when ICG was applied, parathyroid adenomas were detected in all of the patients presenting a high accuracy of this method. Zaidi et al., have shown that the usage of ICG fluorescence angiography enabled the detection of 93% of PGs only by the naked eye [130].

One of the advantages of ICG application is that regarding previous studies, no cases have shown false positive incidents of PGs detection. Furthermore, ICG application is relatively inexpensive and the safety profile for patients is high. This technique is simple for surgeons since instruments used in such surgeries require the same ones as in laparoscopy in which surgeons are usually familiar with. ICG is also not harmful to patients, works quickly, and has a short half-life, which also increases the profile of its lack of toxicity on the organism [76,131]. Regarding ICG limitations, it contains iodine thus, it should not be used in patients with medical conditions that present excessive amounts of iodine in the organism, as well as those who are allergic to this substance. Further, ICG requires a specific source of light and camera filters. Overall, the usage of ICG along with the near-infrared fluorescence seems to be a promising intraoperative technique of detecting parathyroid adenomas and cases of hyperthyroidism [80].

ICG is not the only agent used in the intraoperative detection of PGs. Previous studies have also shown a successful application of aminolaevulinic acid and methylene blue [79,132]. The usage of aminolaevulinic acid provides clear differentiation between PGs and adjacent tissues [133]. Methylene blue has also proved its utility to detect the localization of the parathyroid adenomas with the usage of its low doses [134]. The higher percentage of success (97%) was observed, when higher doses of methylene were applied, which was also associated with the higher risks of cutaneous complications [135].

## 12. Indocyanine Green Fluorescence Vs Parathyroid Autofluorescence

Both techniques—ICG fluorescence and parathyroid autofluorescence—are recent techniques that significantly improve the intraoperative detection of PGs [75]. Regarding ICG application, it is a promising method that provides a higher rate of PGs detection, as well as fewer incidental parathyroidectomies [136,137]. The limitations of this technique include interference from background thyroid fluorescence, which hinders PGs detection, as well as several incidents of false-negative results. Further, there is still insufficient data about the exact relationship between the intraoperative usage of ICG and postoperative hypocalcemia [138,139]. Contrarily, parathyroid autofluorescence does not require the fluorescent dye [131]. In a study performed by Kahramangil and Berber, both techniques—ICG fluorescence and parathyroid autofluorescence—were compared. The results have shown that both methods provide very high rates of PGs detection and a significantly lower number of postoperative hypocalcemia incidents. ICG fluorescence and parathyroid autofluorescence presented similar PGs detection rates equal to 95% (60 of 63 PGs) and 98% (61 of 62 PGs), correspondingly. However, regarding hypocalcemia, parathyroid auto-fluorescence resulted in a higher rate of such incidents (9%) compared to ICG (5%). Even though the utility of both methods in PGs detection is similar, one major difference is the timing needed for PGs detection. Since autofluorescence signal detection is not interfered with by the thyroid gland signal, parathyroid autofluorescence provides PGs detection before the naked eye more often (52% of all PGs) compared to ICG fluorescence (6% of all PGs). In the case of ICG, the dye uptake by thyroid tissue hinders its usefulness in the detection of PGs. ICG requires the administration of an exogenous substance to circulation, which is associated with the risk of allergic reactions. Further, sodium iodine is present in ICG substance, thus, patients with iodine allergy and renal insufficiency are susceptible to harmful effects [77].

## 13. Conclusions

Detection of PGs constitutes a difficult task during surgical procedures within the neck because of a close relationship with important anatomical structures such as thyroid gland, recurrent laryngeal nerve, or inferior thyroid artery. Since additional or ectopic PGs might be present, identification must be as precise as possible, a task which seems to be challenging in both pre- and intraoperative techniques. Despite, all of the preoperative techniques (USG, CT, MIBI) used in PGs detection presents a high accuracy and sensitivity, often there is a need for the application of intraoperative modalities. Intraoperative methods decrease the number of false-negative results, also providing a good approach in the verification of diagnosis. Because of the different properties of each modality, each of them is associated with different advantages and disadvantages (Table 2).

The usage of carbon nanoparticles appears to boost the probability of detecting PGs. As it was presented in a study, a combination of carbon nanoparticles and 99mTc-MIBI significantly facilitated detection of pathological (either adenomatous or hyperplastic) PGs. Raman spectroscopy enables a histological differentiation between hyperplastic and adenomatous PGs; it also provides a specific histological composition of a parathyroid adenoma, which gives information about the prognosis of a patient. Near-infrared autofluorescence proved to be an easily implemented method that is noninvasive, sensitive, specific, and highly facilitates the localization of the PGs. Dynamic optical contrast imaging also uses light radiation to obtain the image of the operative field, although it uses different wavelengths. The image creation is based on fluorescence decay feedback from various tissue fluorophores. The research on this method is in the preliminary stage, hence it is difficult to compare its performance with better explored near-infrared autofluorescence. Also, laser speckle contrast imaging is another non-invasive method used during operations on PGs. Its main usage is the ability to detect microcirculation beneath the superficial layer of tissue. Hence, it may serve as a tool to examine the viability of the glands. ICG seems to be the most promising method in the identification of adenomatous PGs from all the previously aforementioned techniques. In the presented studies, ICG provides detection of nearly 100% of adenomas. Despite some preoperative and intraoperative techniques have been already investigated, further research is needed to determine which modality would be the most satisfying in parathyroid adenomas detection.

## Figures and Tables

**Figure 1 molecules-25-01724-f001:**
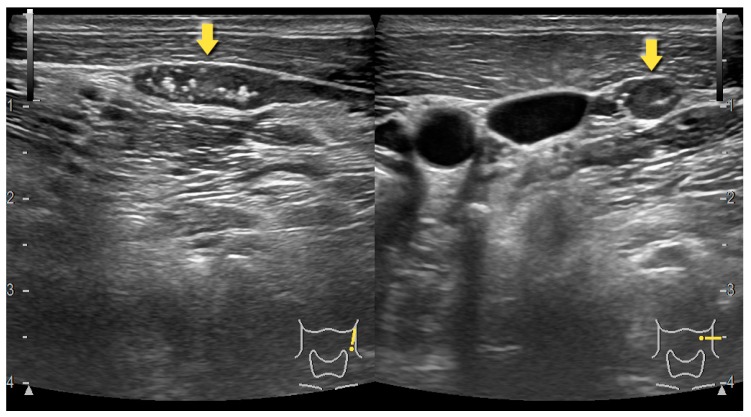
Parathyroid adenoma detected by USG.

**Figure 2 molecules-25-01724-f002:**
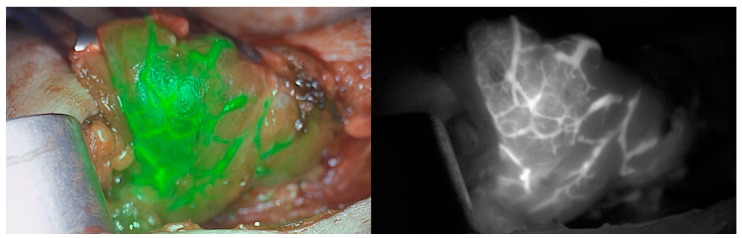
Thyroid gland after ICG application.

**Table 1 molecules-25-01724-t001:** Summary of analyzed papers.

Ref.	Authors	Year	Origin	Method	Usage	No. Patients (Studied Group)	No. Controls (Control Group)	No./% PGs Detected (Studied Group)	No./% PGs Detected (Control Group)	No. Parathyroid Adenomas Detected (Studied Group)	Accuracy	Sensitivity
[43]	Shi et al.	2016	China	Carbon nanoparticles	Intraoperative	52	45	52/52 (100%)	45/45 (100%)	-	-	-
[44]	Yan et al.	2018	China	Carbon nanoparticles with rapid parathyroid hormone detection and ultrasound-guided fine needle aspiration	Preoperative	12	-	-	-	12	Up to 100%	12/12 (100%)
[45]	Chen et al.	2017	China	Carbon nanoparticles & technetium sestamibi (99mTc-MIB)	Preoperative (99mTc-MIB) Intraoperative (carbon nanoparticles)	20	20	160*	-	-	-	-
[46]	Palermo et al.	2017	Italy	Raman spectroscopy	Intraoperative	18	-	-	-	13	100%	-
[47]	Das et al.	2006	UK	Raman spectroscopy	Intraoperative	15**	-	-	-	9	-	95%
[48]	McWade et al.	2014	USA	Near-infrared autofluorescence spectroscopy	Intraoperative	110	6	100%	-	-	-	100%
[49]	McWade et al.	2016	USA	Near-infrared autofluorescence spectroscopy	Intraoperative	137	-	100%*** 98%****	-	-	100%	-
[50]	Kim et al.	2016	Korea	Near-infrared autofluoresccence imaging	Intraoperative	8	-	16/16 (100%)	-	-	100%	100%
[51]	Serra et al.	2019	Portugal	Near-infrared autofluoresccence imaging	Intraoperative	5	-	10/10 (100%)	-	-	-	-
[52]	Benmiloud et al.	2019	France	Near-infrared autofluoresccence imaging	Intraoperative	121	120	390	299	-	-	-
[53]	Paras et al.	2011	USA	Near-infrared autofluoresccence imaging	Intraoperative	21	-	-	-	-	-	-
[54]	McWade et al.	2013	USA	Near-infrared autofluorescence spectroscopy	Intraoperative	45	-	100%	-	-	-	-
[55]	Kim et al.	2017	Korea	Near-infrared autofluoresccence imaging	Intraoperative	38	-	64 92.8%	-	1	92.85%	92.75%
[56]	Falco et al.	2016	Argentina	Near-infrared autofluorescence	Intraoperative	28	-	-	-	9	-	-
[57]	Ladurner et al.	2016	Germnay	Near-infrared autofluoresccence imaging	Intraoperative	25	-	27/35	-	-	-	-
[58]	De Leeuw et al.	2016	France	Near-infrared autofluoresccence imaging	Intraoperative	35	-	81	-	-	-	94.1%
[59]	Squires et al.	2019	USA	Near-infrared autofluoresccence imaging	Intraoperative	59	-	12	-	-	-	87%
[60]	Kose et al.	2019	USA	Near-infrared autofluoresccence imaging	Intraoperative	50	-	192/199 (96%)	-	-	-	-
[61]	Kose et al.	2020	USA	Near-infrared autofluorescence imaging	Intraoperative	310	-	496/503 (98.6%)	-	-	97.6%	98.5%
[62]	Henegan et al.	2019	Australia	Near-infrared autofluorescence imaging	Intraoperative	1	-	-	-	1	-	-
[63]	Alesina et al.	2018	Germany	Near-infrared autofluorescence imaging	Intraoperative	5	-	11	-	1	-	-
[64]	Kahramangil et al.	2018	Argentina	Near-infrared autofluorescence imaging	Intraoperative	210	-	(584/594) 98%	-	-		97-99%*****
[65]	Thomas et al.	2018	USA	Near-infrared autofluorescence imaging + PTeye	Intraoperative	162 (near-IR auto-fluorescence imaging) 35 (PTeye)	-	881 (near-IR auto-fluorescence imaging) 383 (PTeye)	-		92.5% (near-infrared autoluorescene imaging) 96.1% (PTeye)	89.1% (near-infrared autoluorescene imaging) 95.5% (PTeye)
[66]	Kim et al.	2017	USA	Dynamic optical contrast imaging	Ex vivo study Eventually intraoperative	81	-	-	-	-	-	-
[67]	Mannoh et al.	2017	USA	Laser speckle contrast imaging	Intraoperative	20	-	32 (well vascularized PGs) 27 (compromised PGs)	-	-	91.5%	92.6%
[68]	Hattapo ğlu et al.	2015	Turkey	Shear-wave elastography	Preoperative	36	-	-	-	-	-	90% (for parathyroid adenomas)
[69]	Azizi et al.	2016	USA	Shear-wave elastography	Preoparative	57	-	-	-	-	-	-
[70]	Golu et al.	2017	Romania	Shear-wave elastography	Preoparative	22	43	-	-	21	-	93%
[71]	Stangierski et al.	2018	Poland	Shear-wave elastography	Preoperative	65	35	-	-	-	-	-
[72]	Chandramohan et al.	2017	India	Shear-wave elastography	Preoperative	44	-	-	-	39	90.5%	91.1%
[73]	Batur et al.	2015	Turkey	Shear-wave elastography	Preoperative	92	-		-	21	-	85.7%
[74]	Vidal Fortuny et al.	2017	Stwitzerland	Parathyroid angiography with indocyanine green	Intraoperative	73	73	-	-	-	-	-
[75]	Van den Bos	2018	Netherlands	Indocyanine green	Intraoperative	26	-	-	-	-	-	-
[76]	Sound et al.	2015	USA	Indocyanine green	Intraoperative	3	-	-	-	2	-	-
[77]	Kahramangil and Berber	2017 2018	USA China	Parathydoid autofluorescence	Intraoperative	22	-	61/62 (98%)	-		-	-
				Indocyanine green	Intraoperative	3	-	all	-	-	-	-
[78]	Lang et al.	2016	China	Indocyanine green	Intraoperative	70	-	-	-	-	-	-
[79]	DeLong et al.	2017	USA	Indocyanine green	Intraoperative	60	-	60/60 (100%)	-	18/18 (100%)	-	-
[80]	Chakedis et al.	2015	USA	Indocyanine green	Intraoperative	1	-	1	-	1	-	-

- no data; * among 40 patients (studied and control groups combined); ** a total number of parathyroid glands; *** among patients with nontoxic nodular goiter, toxic multinodular goiter, Hashimoto’s thyroiditis, Graves disease, thyroid adenoma, medullary thyroid cancer, primary hyperparathyroidism; **** among patients with differentiated thyroid cancer; ***** a range of sensitivity % among three centers.

**Table 2 molecules-25-01724-t002:** Advantages and disadvantages of intraoperative parathyroid detection methods.

Method	Advantages	Disadvantages
**Dynamic Optical Contrast Imaging**	Non-invasive No admission of exogenous substances Instant feedback	Not enough evidence
**Laser Speckle Contrast Imaging**	No admission of exogenous substances Instant feedback Assess of viability of PGs	High susceptibility to movement of the operation field
**Autofluorescence Spectroscopy**	Non-invasive No admission of exogenous substances; Instant feedback	No information about the viability of PGs Requires the blackout of the operating room light Limited ability to localize PGs covered deeply by other tissues
**Autofluorescence Imaging**	Non-invasive No admission of exogenous substance Instant feedback Contactless Possibility to differentiate adenomas Possibility to display on the operation field	No information about viability of PGs Requires the blackout of the operating room Limited ability to localize PGs covered deeply by other tissues
**Raman Spectroscopy**	Non-invasive No admission of exogenous substance; Possibility to differentiate adenomas	Requires additional time
**Carbon Nanoparticles**	Do not penetrate to tissues Visible in the operation field	Admission of exogenous substance Requires precise injection Do not differentiate adenomas
**Shear Wave Elastography**	Non-invasive No admission of exogenous substance; Possibility to differentiate adenomas.	Dimensions of parathyroid adenomas cannot be estimated
**Indocyanine Green**	Inexpensive Safe Possibility to differentiate adenomas Visible in the operation field	Admission of exogenous substance Contrast nonspecific for PGs Contains iodine

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
