# Peer review of "Preoperative and Intraoperative Methods of Parathyroid Gland Localization and the Diagnosis of Parathyroid Adenomas"

_molecules, 2020, doi:10.3390/molecules25071724_

Round 1

Reviewer 1 Report

The work is well prepared and recapitulate the current available tools to distinguish, visualize, discover parathyroid glands.

My suggestions are below:

  • please correct numerous English expressions/styles through all the manuscript.
  • summarize the manuscript, some sentences are redundant and weigh down the reading

Author Response

Jacek Baj

Medical University of Lublin

Department of Human Anatomy

Jaczewskiego 4

20-090 Lublin, Poland

[email protected]

3rd March 2020

Dear Reviewer,

Thank you very much for reviewing our manuscript. We are very grateful for the commitment you have provided during the revision of this manuscript and appreciate your precious comments.

We are pleased to submit the manuscript entitled ‘Preoperative and intraoperative methods of parathyroid glands localization and the diagnosis of parathyroid adenomas’.

            The manuscript has been checked once again in terms of English by one of our native speaker friends and we hope that now it is much more improved. We have also deleted several irrelevant information throughout the text and hope that now it is clearer and more readable. We have added several important information in some sentences in the text.

We would like to thank you again for your effort, feedback, and extremely helpful comments. We hope that now the manuscript is improved.

We wish you all the best!

Sincerely,

Jacek Baj,

on behalf of the authors

Reviewer 2 Report

This is a review study on pre-and intra-operative methods of parathyroid glands localization for parathyroid adenoma surgery. This review focused on application of several technique to detection of parathyroid adenomas. Although this article has well-summarized several studies of this topic, it has some limitations.

  1. The main one is that this review is not a systematic review and, consequently, the level of evidence is reduced. A systematic review and meta-analysis (following the methodology established by PRISMA) that summarize the diagnostic values of various techniques, might be interesting.

  1. This article does not contain the roles of USG or 99mTc-MIBI or 4D CT, which have been widely used for preoperative localization. The authors could elaborate further their discussion on application of other various techniques, comparing to “conventional” techniques such as USG or 99mTc-MIBI or 4D CT.

  1. This article enumerates various techniques like a list. The authors might summarize the advantages and disadvantages of techniques they discussed in a table format.

  1. In addition, each technique or each reagent should be more specifically discussed. For example, what is carbon nanoparticles? What are the major chemical structures of them? Why do not the authors try to employ figure or illustration to explain them?

Author Response

Jacek Baj

Medical University of Lublin

Department of Human Anatomy

Jaczewskiego 4

20-090 Lublin, Poland

[email protected]

3rd April 2020

Dear Reviewer,

Thank you very much for reviewing our manuscript. We appreciate the interest and commitment you have provided for this work. We are very grateful for your extremely precious comments. We are convinced that thanks to your suggestions this manuscript will be much more valuable.

We are pleased to submit explanations and details of our revisions in the manuscript entitled ‘Preoperative and intraoperative methods of parathyroid glands localization and the diagnosis of parathyroid adenomas’.

The followings are our point-by-point responses:

  1. To increase the quality of this manuscript, we have added a separate paragraph ‘Materials and methods’ to show the strategy of research. Further, in the same paragraph, we have added a table that summarizes the studies described in this paper. In a table, we have included the following information (wherever possible): authors, year of a study/publication, origin, a method which was used in a study, number of patients and control groups, number of detected parathyroid glands and/or parathyroid adenomas, accuracy, and sensitivity. We hope that this will facilitate the reading and understanding of the value of each study described in this paper.
  2. We have added a separate paragraph about conventional preoperative imaging techniques (ultrasonography, computed tomography, and sestamibi scintigraphy) and their utility in parathyroid glands and/or parathyroid adenomas detection.
  3. We have added a table, which summarizes all of the intraoperative modalities described in this paper in terms of their advantages and disadvantages.
  4. We have added a short subparagraph, which describes carbon nanoparticles generally.

We have also deleted several irrelevant information throughout the text to make it clearer and more readable. We have also added several important information in some sentences in the text. Also, the whole manuscript has been checked in terms of English by one of our native-speaking colleagues.

We would like to thank you again for your effort, feedback, and extremely helpful comments. We hope that now the manuscript is improved.

We wish you all the best!

Sincerely,

Jacek Baj,

on behalf of the authors

Reviewer 3 Report

This is a non-systematic review of literature by Baj et al that examined published studies reported on a selected preoperative and intraoperative methods used in localizing the parathyroid glands. The study is well written and organized. The study provides a good summary of the advantage and disadvantages of the described methods.

Overall, the study excels in providing a detailed description of the mechanism and the process of each methods, however, it falls short in providing similar details regarding the clinical utility of those methods in few places. for instance, for the “Shear wave elastography”, the authors are citing some studies and using general term such as “more accurate, or significant difference” without providing data regarding the difference in accuracy or whether the statistically significant difference is clinically meaningful.

The other concern is the non-systematic nature in including studies in this review. The included studies are of various level of evidence and in few places the authors are not making that clear. For instance, the authors cited one study that evaluated the effectiveness of carbon nanoparticle (Shi et al) and passing their conclusion without providing a background on the study, furthermore they passed part but not whole conclusion, reviewing the Shi et al study specifically demonstrate that the parathyroid gland was identified 100% in both the treatment and control groups.   Similar pitfalls repeated throughout the paper, my suggestion is if the authors are not conducting a systemic review about the quality of data retrieved from literature, there should be a clarification with each cited paper regarding the quality of evidence and whether the outcomes were reproducible.  

Lines 63-65: The stated risk of reoperation is rather high (or likely represent an upper limit), multiple factors effect the risk of reoperation, some centers have reported a risk as low <5%.

Line 120: How was nanocarbon particles used pre-operatively? It is intraoperative method.

Line 436: typo “an easily implanted method”.

Author Response

Jacek Baj

Medical University of Lublin

Department of Human Anatomy

Jaczewskiego 4

20-090 Lublin, Poland

[email protected]

3rd April 2020

Dear Reviewer,

Thank you very much for reviewing our manuscript. We appreciate the interest and commitment you have provided for this work. We are very grateful for your extremely precious comments. We are convinced that thanks to your suggestions this manuscript will be much more valuable.

We are pleased to submit explanations and details of our revisions in the manuscript entitled ‘Preoperative and intraoperative methods of parathyroid glands localization and the diagnosis of parathyroid adenomas’.

The followings are our point-by-point responses:

  1. We have added several important information in those sentences, which were describing a utility of a particular modality to specify it. Further, we have also added a table, which summarizes the details of the studies described in this paper. Another table was also added and this one includes the advantages and disadvantages of specific intraoperative methods described in this paper.
  2. Lines 63-65: We have clarified this sentence by stating that 30% constitutes rather an upper limit of reoperations and added information that several centers state that the risk is rather low (below 5%).
  3. Line 120: We are sorry for this mistake. We have corrected it for “intraoperative”.
  4. Line 436: The typo has been corrected.

We have also deleted several irrelevant information throughout the text to make it clearer and more readable. We have also added several important information in some sentences in the text. Also, the whole manuscript has been checked in terms of English by one of our native-speaking colleagues.

We would like to thank you again for your effort, feedback, and extremely helpful comments. We hope that now the manuscript is improved.

We wish you all the best!

Sincerely,

Jacek Baj,

on behalf of the authors

Round 2

Reviewer 2 Report

All comments have been addressed

Reviewer 3 Report

The authors have addressed my comments sufficiently.